# Visualising quantum innovation: A regional case study

**O. Jiménez Farías**[ID]◉*, **Arnau Demergasso**◉, **Maryam Vaziri**◉, **Sergi Vives Rodón**◉, **Nelly Canessa**◉, **Eoín Phillips**[ID]◉*

Smart Society Research Group, La Salle-Ramon Llull University, Barcelona, Spain

◉ These authors contributed equally to this work.
* osvaldo.jimenez@salle.url.edu (OJF); eoinedward.phillips@salle.url.edu (EP)

## Abstract

At the beginning of this century, the advent of a second generation of 'quantum technologies' was announced together with its revolutionary potential to change existing information technologies. Despite the rapidly increasing paid to quantum technological development, there has been little attention paid to the specific characteristics or relationships within emerging quantum ecosystems. The aim of this study is to visualize the innovation structures and relationships that are emerging to shape these technological developments. As these structures typically depend on specific regional features, we have specifically focused on the Spanish case, as it is potentially indicative of the differences between European innovation models and other regional patterns. This objective was achieved by researching the funding network of the ecosystem, collected from a systematic review of various official sources and relevant previous literature. The resulting dataset was framed using the Innovation Ecosystem model and broken down through network analysis theory, as well as characterized through descriptive statistics. This framework identified the significant role that projects play in European scientific and technological innovation, which work as hubs to concentrate resources and incentive cooperation between actors. This is relevant because current work on quantum technologies neglects their importance, as their analysis focuses on the quantity of institutions rather than their relations. Moreover, this paper points out the prominence of public funding to drive quantum innovation, largely stemming from the European Commission. This is another key mechanism that is missed by the existing literature. Finally, it also sheds light on the recipients of this funding, who are mostly research centres. These results allow us to conceptualize the Spanish quantum ecosystem and offer the opportunity for comparative studies with other quantum technologies ecosystems.

## Section I: Introduction

### Historical background

Quantum technology stands for the set of artifacts and techniques designed to complement or supplement existing information and communication technology based on unprecedented control of microscopic systems [1]. In 2016, the Chinese space agency launched the satellite

**Data Availability Statement:** All relevant data are within the manuscript and its Supporting Information files.

**Funding:** O.Jiménez Farías (Recipient).: Ajuts a l'activitat de recerca del personla docent i

investigador de la Universitat Ramon Llull, 2023-URL-Proj-078 Eoin Phillips (Recipient): Grant PID2019-105131GB-I00 from the Spanish Ministry of Science and Innovation. The funders had no role in study design, data collection and analysis, decision to publish, or preparation of the manuscript.

**Competing interests:** The authors have declared that no competing interests exist.

Micius devoted solely to quantum communications purposes [2]. Inside the satellite, a source of quantum light sends beams of entangled photons to two distant locations on Earth with the aim to provide communications channels secure from hacking attempts at distances longer than any existing system on Earth. In 2019, Google announced the first 'speedup' achieved by a quantum computer over any other existing computer system [3]. These two milestones in the field of quantum technology attracted the interest of most and announced an era of intense technological development that extends around the globe [4]. Potential applications of quantum technology include drug discovery [5], financial modelling [6], weather forecasting [7], cybersecurity [8], machine learning algorithms [9], and blockchain [10].

In the European context, Spain is one of the six nations that will allocate a quantum computer as part of the program EuroHPC JU together with Germany, Poland, Italy, Czech Republic, and France [11]. A conglomerate of start-ups purely focused on developing or applying quantum technologies have emerged, and these have been joined by existing companies, research institutes, and universities who are taking position and declaring themselves part of the Spanish quantum ecosystem.

## Objective of the study

Despite the longstanding conceptualization of quantum technologies as a distinct technological aim (and therefore as a distinct innovation process), the specific institutional structures and relations that have either given rise to the development of quantum technologies or which are emerging in the pursuit of these innovations have received very little scholarly attention. It has been shown in literature of technological innovations and scientific practice that the institutional structures that give rise to innovations are rarely universal. Rather, the specific structural organization of innovation institutions may depend upon specific social, economic, and political features of the region in question [12].

To put it another way, there are a range of ways in which the development of technologies may come about amongst which it is common to identify vertical, horizontal, triple helix [13] or ecosystem types of institutional arrangements [14]. It is the assumption of this article that the same must be true of the emergence of quantum technologies and quantum computers; meaning that it may be possible that quantum technologies in different regions may be brought about through different institutional/organizational arrangements.

As the perspectives of technological change are big, we recognize the relevance of consistently considering the characteristics of the actors, their origins, the role they aim to play, and how they relate to the other actors in the network. To our knowledge, this work represents the first academic work that attempts to capture the mechanisms that drive such a quantum ecosystem. By a systematic collection of data available in official channels and declarations, we have recreated the funding network of the Spanish ecosystem. The relation 'funding' is just one aspect of the ecosystem but it is a necessary one if we accept the fact that the state of the innovation ecosystem in Spain is primal. Given the complexity of the ecosystem even in this early state, we assume our description cannot be exhaustive but we argue it is sufficient to describe basic properties inherent to its emergence that will impact its evolution.

## Structure of the study

In Section I, we present a literature review that considers a set of reports by several organizations like the World Economic Forum, Ametic, McKinsey, Quantum Flagship, IBM, IQM, among others whose purpose is to promote awareness of the state of the art in the development of quantum technology. Interestingly, most of the reports number a set of institutions or collaborators that could be considered 'self-named' players in the quantum ecosystems—though

the type of collaboration that links institutions within these ecosystems are not always evident. We complement this review with state-of-the-art literature on innovation ecosystems from where we conclude an important kind of actor must be considered in the description of the network: the European projects.

Section II describes our method of data collection that is based on the information that the actors make available. Different attributes such as kind of institution and the type of technology they are concerned with are recorded. We observe that the market relation producer-user is not well determined at this stage of the ecosystem but funding relation between among actors is more commonly declared. In some cases it is possible to track the exact amount of money that is involved in the relation but as it is not always the case, we restrict our analysis to the relation receiver-provider between them.

Network analysis is shown to be a useful description of the ecosystem revealing a rich structure of interactions as well as the most prominent actors within it. Visualization of the funding network as well as the most common measures in networks theory are presented in section III following descriptive statistics from the gathered data.

A discussion of our methods and data base is presented in Section IV where we discourse on possible ways in which our analysis can be extrapolated to the European region and is not exclusive to the Spanish case.

Finally in section V we present our concluding remarks.

### Literature review and limitations of current studies.

Over the past twenty years, we have been told that we have experienced one, two or even three quantum revolutions [15]. Whilst this temporal dimension of technical change has been stated—and the possibility of debate of the extend of this change has been opened up—rather less has been said about the spatial dimensions of such change; that is what is the relationship between the character of quantum technological development and the places in which such development is said to be taking place.

Existing literature on the state of quantum technologies in Spain may be understood as falling under two types of publications. On the one hand, we have state-of-the-art reports produced in large by private consulting companies [16–19]. On the other hand, we have several academic articles [1, 20]. In this section, we will detail the different approaches taken by the two strands of literature, offering a non-exhaustive sample of the most relevant reports and papers, as well as detailing the gaps and limitations of their methodology (see S1 Table).

Over the past 3 years, 15 reports have been published related seeking to describe the level of activity related to quantum technological innovation. In general, the regional focus of such reports is extremely broad. Aside from those that focus on the EU or the US in general [4, 21], most of them try to describe the industry on a global scale [17–19]. Only one report [16] has focused on the Spanish case (made by the Spanish consulting agency AMETIC), thus making any cross-referencing and comparison impossible.

According to AMETIC as of 2023, there were 67 institutions or firms that should be regarded as being included in the Spanish quantum technological development. The criterion for inclusion is based on 'own wording' [16] and includes companies, R&D centers, national projects/initiatives, and regional partnerships/ecosystems. However, in its report, each institution is listed sequentially with no sense of relation between them (see Fig 1).

Like AMETIC's Spanish report, the other industry reports also represent quantum technological activity with regards to particular kinds of output: patents, peer-reviewed articles, total funding allocated or total number of players. However, they also follow AMETIC's logic in representing their chosen institutions sequentially rather than relationally. As such, the reports give little indication of the ways in which institutions connect to each other except presumably as individual agents competing against each other across large regional areas for similar ends.

| Name | Size | Description |
|---|---|---|
| DAS Photonics | SME | Platform integrators in the defence, avionics and space industries. Spin-off from the Nanophotonics Technology Centre at the Polytechnic University of Valencia. They participate in the CUCO project. https://www.dasphotonics.com/ |
| Entanglement Partners SL | SME | First quantum consulting company created in Spain and Latin America. Its main activity focuses on strategic and technological consultancy relating to quantum technologies. https://www.entanglementpartners.com/ |
| G2-Zero | SME | Start-up created by IMN-CNM (CSIC) researchers to manufacture single photon sources, with applications in communication and quantum technologies. https://g2-zero.com/ |
| Inspiration-Q | SME | Technology-based company created from the CSIC. It designs and markets quantum and quantum-inspired solutions for optimisation, simulation and machine learning problems. https://www.inspiration-q.com |
| iPronics | SME | It develops programmable photonic integrated circuits for all layers of industry. Photonic processing for greener hardware in communications, sensor and computing applications. https://ipronics.com |
| IQM | SME | Leading pan-European quantum computer company with its headquarters in Espoo, Finland. It has recently opened a subsidiary in Spain (Bilbao), focusing on quantum finance and co-designing quantum computers. https://www.meetiqm.com/ |
| LuxQuanta | SME | LuxQuanta is an ICFO spin-off based in Barcelona. Focused on providing quantum key distribution (QKD) systems and technologies for integration into existing network infrastructures. https://www.luxquanta.com/ |

| Name | Size | Description |
|---|---|---|
| Multiverse Computing | SME | European leader in quantum computing. With 70 employees in San Sebastian, Toronto, Paris and Munich. They have a portfolio of over 30 patents. They have their own product: Singularity. This is dedicated to applying quantum computing and "Quantum Inspiration" to problems in different fields of application: finance, economics, aerospace, health, automotive, industry 4.0, logistics, etc. https://multiversecomputing.com/ |
| Qcentroid | SME | It is the first company to deliver Quantum technology to Web3 ecosystems; offering Quantum capabilities, HW and algorithms to Web3 organisations and projects. It provides quick, easy access to quantum algorithms. https://qcentroid.xyz/ |
| Qilimanjaro | SME | They design and market annealer-type quantum computers. They manufacture the complete stack: quantum chip, control software and development libraries. https://www.qilimanjaro.tech/ |
| Quantum Mads | SME | They offer the hybrid QSaaS tool that enables their customers to tackle the most challenging industrial problems. Its aim is to dissect the intrinsic dynamics of complex industrial systems and create innovative, hardware-independent solutions. https://quantum-mads.com/ |
| Quanvia | SME | Focused on implementing quantum computing applications and opening up the spectrum of quantum computing to a wider audience. They offer research, consultancy and training services https://www.quanvia.com/ |
| Qurv | SME | Spin-off of ICFO. They manufacture wide-spectrum image sensors. Their sensors are based on quantum dot (or well) technology. This technology enables signals from the visible to the short-wave infrared range to be detected, and can be integrated with today's low-cost, high-end CMOS sensors. https://www.qurv.tech/ |

**Fig 1. Extract from AMETIC's report listing companies in the Spanish quantum ecosystem.**

Unfortunately, the academic literature also does little to shed light on the quantum industry in Spain. This is because there is no academic paper that focuses on the region. Moreover, the existing literature tends to focus on specific aspects of the industry—such as the number of start-ups, patents, or job expectations—and does not help to comprehend the structures of quantum technological development [20, 22].

Nevertheless, there is a common thread between the existing literature which is their rhetorical approach to the quantum technologies industry. The reports talk about an 'industrial ecosystem of quantum technologies' [16], a 'new computation quantum ecosystem' [23], a 'national quantum ecosystem' [18], a 'QC ecosystem' [19], or a 'quantum communications ecosystem' [17]. The academic articles also follow this rhetoric exemplified by formulations such as 'the landscape of the quantum start-up ecosystem' [22].

Despite the rather loose use of the term ecosystem in these articles and reports—it is unclear in these accounts what precisely accounts for an ecosystem rather than any other form of relation between institutions, companies, and universities—the rhetorical use of the term is clearly doing a degree of work to make sense of the range of institutions involved in the development and deployment of quantum technologies.

## Conceptual framework

To further understand the use of the term ecosystem to describe the character of quantum technological development in Spain and to better analyze the forms of relations between the institutions involved in its development, this article suggests following an 'innovation ecosystem' perspective. This perspective not only looks 'at the type and number of actors but focuses on the types of relationships between them' [14]. In other words, the ecosystem approach

focuses on the 'network of interdependent actors who combine specialized yet complementary resources and/or capabilities' [24] to produce any technological innovation.

Additionally, this technique of mapping out the 'collaborative arrangements' [25] captures the set of shared assets, standards, and interfaces that underpins an activity system around any technological innovation [26] and stresses its 'multilayer' nature [27]. In this sense, the ecosystem's approach takes the individual firms as intrinsically part of a network of cooperation and competition with other firms [28], thus making the network the principal unit of analysis [29]. Moreover, it allows for an examination of both the macro level of national policy and the micro-level of individual firm activities [14]. In summary, this approach understands that the performance of one actor influences the performance of others and that of the entire innovation ecosystem [25].

This has a clear implication for the literature behind quantum technologies. On one hand, the relevant reports and articles declare the quantum technologies industry as an ecosystem. However, they do not demonstrate any structural network behind the activities, actors, and locations. In short, the existing literature uses a certain rhetoric with a clear theoretical background without actually employing it.

In contrast to existing literature on quantum technological development, the approach taken by this article allows us to specifically characterize the type of network that is shaping quantum innovation, giving a holistic picture of its wider mechanisms and general trends. This approach will permit a specific analysis of the Spanish case and its geographical particularities, which will be indicative of the innovation patterns used in the European Union (and warrant further comparison with other regions). Finally, by establishing concrete links between actors, we will assess the issue in an unbiased manner rather than trusting internal sources.

It's important to highlight the concept of 'actors' as they represent the individual units connected by the links within the network. This concept is loosely defined by the ecosystem literature as exemplified by Adner's [25] definition of actors being the 'entities that undertake the activities' in the ecosystem. Consequently, actors will be considered as organizations who possess agency within the ecosystem to in some way affect or contribute to its growth as opposed to 'factors' (such as public policy) which refer to wider social and economic structures [30].

This is relevant because one of the key actors found by our study are the European (and to a less extent national) projects which are neglected by the rest of the academic literature and private sector reports. For instance, AMETIC only names 9 projects while we found 54 of them (see Section III). This oversight may be caused by the 'untraditional' nature of the projects as they are not firms, governmental organizations, research centres, or other more established types of institutions. Nevertheless, they are actors precisely because they possess a particular agenda (both in terms of time and objectives) that is aimed at creating quantum technological innovation and thus affects the growth of the wider ecosystem.

## Section II: Research design (approach)

### Data collection and sample

To carry out our research objective of understanding the mechanisms driving quantum technological innovation in Spain, we have collected a broad sample of the actors in the Spanish quantum ecosystem. This will allow for comprehension of each actor's characteristics, origins, the role they play, and connections with the other actors in the network, thus allowing for an analysis of the mechanisms behind quantum innovation.

This sample was collected through two types of secondary sources: (1) all the projects, ecosystems, companies, and research centres declared by AMETIC in their 2023 report presenting

**Table 1. Total actors.**

| Sample Size based on the research criteria | |
|---|---|
| Actors declared by Internet searches | 154 |
| Actors declared by Ametic's report | 67 |
| **Total number of actors** | **221** |

the state of the quantum industry in Spain [16]. (2) All the entities disclosing on the Internet that they receive and/or provide funding for working with quantum technologies in Spain. In other words, all the entities that fall under one or both these categories are declared as potential actors in the Spanish quantum ecosystem and are included in our sample. We have identified 67 actors with the first source and 154 with the second one making up 221 actors in total (see Table 1). For a comprehensive breakdown of all the actors' names and their sources, consult S2 Table.

Identifying the actors according to the first criterion was simple since AMETIC's report provides a comprehensive list of these companies, research centres, projects, and ecosystems. We have taken it exactly as it is presented. However, the second criterion is more complex since there are several sources through which we have accessed the funding schemes of the actors. In other words, there is no unified source that provides all the entities receiving or providing funding for quantum innovation in Spain.

Consequently, this information was tracked though internet searches which has led us to several more sources as shown in Table 2. First, the European Commission has the CORDIS database which declares every project under their funding schemes. Therefore, we have selected the projects that work on quantum technologies while containing one or more Spanish institutions and added them to our sample. The total number of actors found in this database is 48. Second, some actors come from official documents from the Spanish Government or related agencies detailing their funding schemes. These make up 45 of the total number of actors. Third, 51 actors declare their funding sources on their own webpages. Finally, a few outliers have been tracked through news sources or other European agencies which account for the remaining 7 and 3 actors respectively.

In summary, we have used these two sources (AMETIC and Internet searches) to compile a database of every actor participating in the Spanish quantum ecosystem. For the second source, further tracking through European databases, official government documents, webpages, and news sources was warranted. However, compiling the sample was only the first part of our methodology. Once the raw data was gathered, we allocated three different variables to each actor which helps in understanding their relationships, roles, and activities in the ecosystem.

## Studied variables

While there are many potential relationships that may be explored, this paper has focused on the stated financial relationship between actors referred to as 'funding role'. Consequently, by

**Table 2. Actors found through Internet searches.**

| Sample Size based on the research criteria | |
|---|---|
| Actors declared by the CORDIS database (either as projects or participant entities) | 48 |
| Actors declared by the Spanish government or related agencies | 45 |
| Actors declared on webpages | 51 |
| Actors declared on news sources | 7 |
| Actors declared by the EDF | 3 |
| **Total number of actors** | **154** |

establishing their funding schemes, a certain type of relationship between actors will be defined which can be used to infer some properties of the ecosystem (see Section IV).

Therefore, we have classified every actor under the following criteria: (1) Providers (2) Receivers (3) Both (4) Not Specified. These titles refer to the directionality of the funding meaning if they receive or provide funding for quantum technologies or do both simultaneously. Hence the actors were first distributed according to their funding role which was subdivided into providing, receiving, doing both simultaneously, and not specifying it.

Additionally, the role and activities of the actors in the ecosystem have also been categorized according to broad innovation types. The actors were classified by quantum technology field meaning which area of the industry they state they are working in or for. Such fields were (1) Computing (2) Cryptography (3) Communication (4) Sensing (5) Simulation and (6) Chemistry. Note that one actor may be working in more than one of these fields.

Finally, the last established variable was the type of actor. Consequently, the actors can also be (1) University Research Centres (2) Startups (3) SMEs (4) Large Companies (5) Governments (6) Independent Research Centres (7) Banks (8) Projects and (9) Hubs. This variable thus determines the nature of the actors and helps in comprehending their role inside the ecosystem as well as further characterizing its funding relationships (see Section IV). A detailed classification of every actor under these three variables is also offered on S2 Table.

There is one important consideration to be made about this methodology. As the reader may have already guessed, this is not an exhaustive method meaning that our database contains only some possible actors under the set characteristics. In other words, some entities with funding schemes for quantum technologies do not appear on the list simply because some institutions do not declare them. Moreover, we may have missed some that make their funding schemes public since cross-referencing our criteria with every Spanish company and institution is impossible.

Yet this lack of exhaustivity does not invalidate our methodology. Since our overall aim is to provide a general picture of the ecosystem's character (rather than a list of all its actors), having some gaps in the number of actors is acceptable given our objective. As shown in section III and IV, we are confident that we have found its key actors which provide the ecosystem's general trends. Therefore, although not exhaustive, the methodology chosen perfectly serves our aims thus validating its use in this paper.

## Section III: Results

### Descriptive statistics

As stated above, an initial objective of the project was to determine the different areas that the actors report to be working in. First, the data shows that 32% of the total number of actors are involved in developing Q-Computing technologies as illustrated in Fig 3. Thus, Q-Computing is the most populated field. Second, Q-Communication technologies contain 20% of the total actors. Third, Q-Sensing and Q-Cryptography are tied with 14% of the actors respectively. Finally, the more niche fields of Q-Simulation and Q-Chemistry only involve 10% each. An exhaustive breakdown of this variable is displayed in Figs 2 and 3.

Another defined variable was the type of actor which indicates their nature. According to our data, the most common actors are the projects. This is because they make up 55 out of the 221 total actors (see Fig 4) or an even 25% as illustrated by Fig 5. SMEs follow with 20% of the total share but Large Companies only account for 8%. Moreover, Research Centres constitute 23% of the ecosystem but startups are limited to 7% of the split. Finally, Hubs and Governments form 8% and 7% while Banks make up only 2% of the actors.

However, our principal aim was to establish funding relations between the actors in the ecosystem. Therefore, this section will present the results of our research within the parameters

Q-Technology Field total count

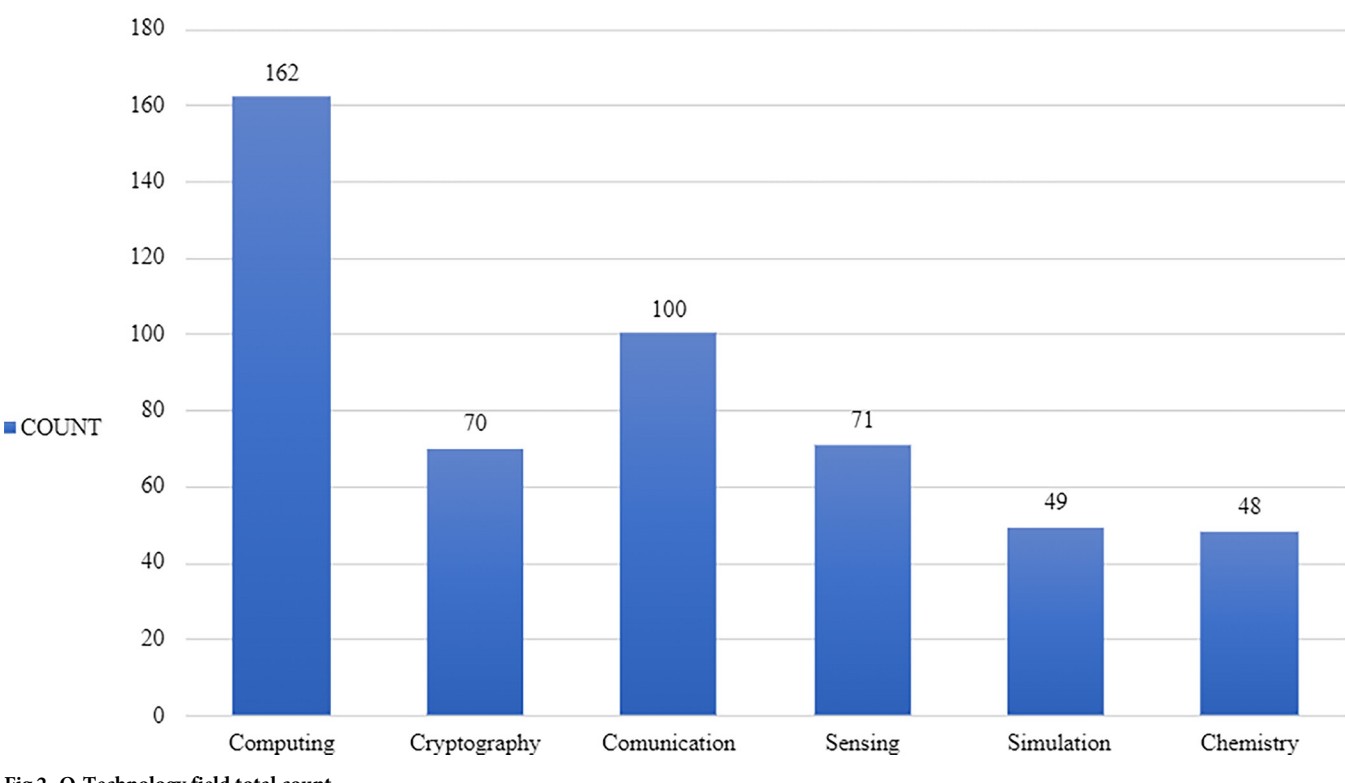

**Fig 2. Q-Technology field total count.**

presented in Section II which is to say the split between Provider, Receiver, Both, and Non-Specified actors. Furthermore, these descriptive statistics will be the backbone of the following network analysis.

To begin with, as shown in Fig 6, most of the actors in the Spanish ecosystem fall under the category of Receivers. What stands out in the graph is that the number of Receivers barely falls short of half the total number of actors. Concretely, 109 actors out of the 221 total ones exclusively receive funding. This is followed by the Both category which gathers 60 actors and the NS actors which amount to 30. In contrast, the Provider category only numbers 22 actors (see Fig 6).

This imbalance is further illustrated by Fig 7 which details the split between categories in terms of percentage. Again, Receivers account for 49.3% of the total network compared to the 10% of pure Funders in the ecosystem. The gap is worsened if we consider that the category Both is the second most numerous amounting to 27.1%. This is significant because the actors in that category simply transfer funding between institutions and do not generate funding from within themselves.

This recurrence of the Both category is interesting because it displays that one-quarter of the total actors act simply as intermediaries between Funder and Receiver institutions. We will discuss later the nature of these actors in Section IV. Suffice to say for now that there is a clear increase in the number of actors going from the Provider to the Receiver categories which is mediated by the Both categories. In short, Figs 6 and 7 show that there are few Funders and many Receivers in the Spanish quantum ecosystem.

## Q-Technology Field by percentage

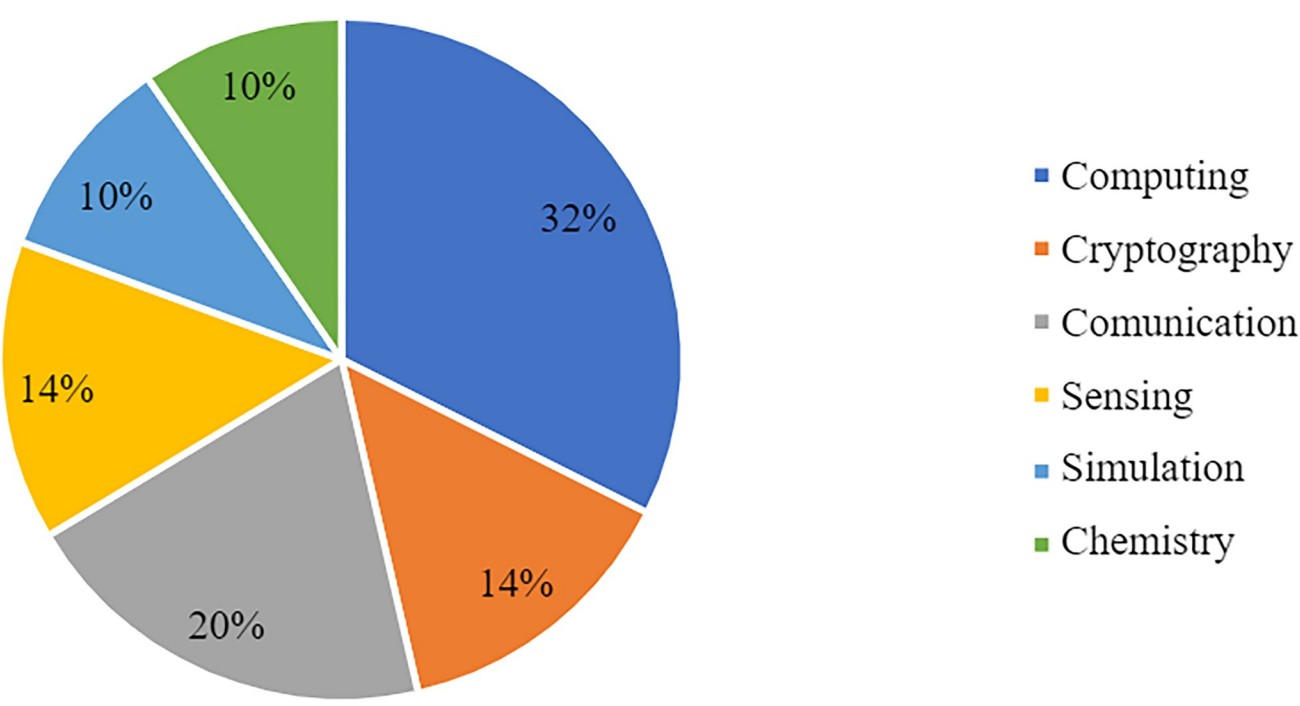

**Fig 3. Q-Technology field by percentage.**

Finally, another remarkable outcome of our research is the high percentage of NS actors in the network which adds up to 13.6%. This, as justified in Section II, does not indicate a methodological failure but rather a feature of the ecosystem, a lack of transparency and a tendency of many actors to not declare their activities. Moreover, this grants credence to further investigation around these actors given that one tenth of the total ecosystem does not provide clear indicators of their role in it (since it is not involved in any of the projects or initiatives found by our investigation).

To conclude, we have stated that the most numerous type of actor are the projects. Therefore, further analysis of the funding sources for this particular type of actor is warranted. Again, the investigation has shown that of these 55 projects, 34 of them receive funding from the European Commission which accounts for 62% (see Fig 8). In contrast, the Spanish Government, the Generalitat, and the Basque Government (all national institutions) combined only fund 36% of the projects as they are responsible for funding 8, 7, and 2 respectively (plus 3 that combine Catalan and Spanish funding).

### Network visualizations

To describe the emerging quantum ecosystem and its development, we use a network analysis approach based on the relation receiver-provider (links) between the different institutions or

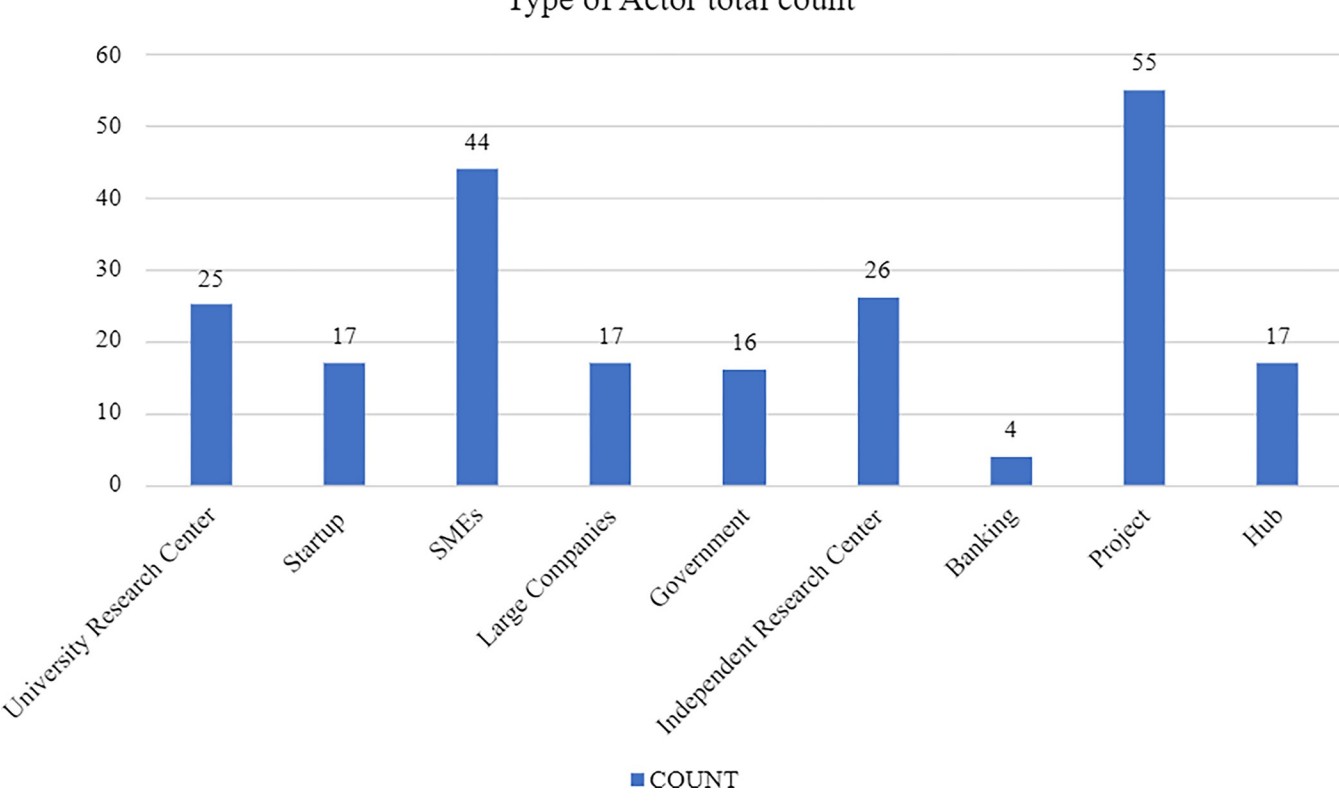

**Fig 4. Type of actor total count.**

actors (nodes). In this representation, each node has a funding role that characterizes it as provider (P), receiver (R), receiver and provider (RP), and non-specified (NS). The funding roles will be represented by the colours in Fig 9. Thus yellow will be NS, blue will represent P, purple will stand for R, and green for RP.

An arrow emerges from a P or an RP node, whereas R nodes are connected to others only through the tip of an arrow giving the structure of a directed graph. The NS state of a node indicates that its funding relation with any other actor could not be identified by the application of our methodology, hence it appears as an isolated node in the network. The network consists of 221 nodes and 297 edges as represented in Fig 10a. For a first imaging of the network, we have chosen a circular layout where the main P nodes appear at the centre of an inner circle of PR nodes and an external circle of R nodes and the remaining NS nodes in peripheral regions.

In Fig 10A, we present a visualization of the Spanish quantum ecosystem where all the nodes have the same size but colour differentiates their funding status (see Fig 9). In Fig 10B, nodes are sized according to their total degree and highlighted those that have a degree bigger equal than 2. Fig 10C and 10D show a similar representation as Fig 10B but with In-degree and out-degree respectively.

## Degree of centrality

We describe statistical features of the network by their degree of centrality which measures how connected a node is. Since this is a directed graph, we must differentiate between three types: (1) The total degree accounts for the total number of edges for each node. (2) The In-

## Type of Actor by percentage

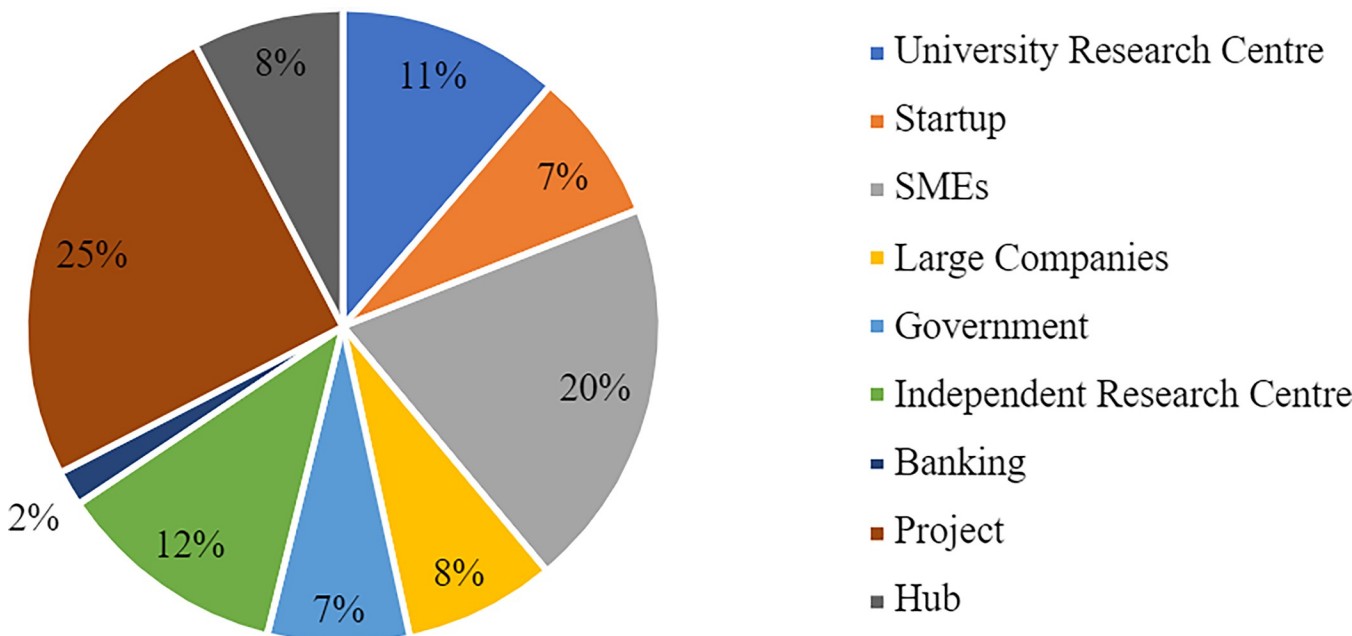

**Fig 5.  Type of actor by percentage.**

degree represents the number of incoming relations meaning how many tips of arrow edges a node receives. (3) The Out-degree orders the nodes according to the number of tails of arrow edges and therefore a node with high out-degree indicates a more active provider in the network.

As it is observed in Fig 11, histograms in all cases are skewed to the left showing that in this initial stage of the Q-ecosystem most of nodes have very few connections. Total degree plot presents a maximum at value 1 meaning that most of the nodes have only one interaction of any kind with the rest of the networks. In-degree have a maximum at value 1 saying that most of the nodes have only one source of funding. Out-degree has a maximum at 0 value reflecting the fact that most of the nodes do not provide any resources. Given the observed sparseness of the network and for the purposes of visualization in Fig 10B-10D, we weight nodes according to its degree.

At the tails of the distributions, we found the highest values of centrality and hence the most interacting actors in this network. To identify the most active nodes in the network, we look for the percentage of actors that have centrality values bigger equal than 2 which for Total degree it is 46.6%, In-degree is 21.2%, and Out-degree is 19.5%.

## Section IV: Discussion

The data presented above allows for several structural inferences to be made. Once the funding links between the ecosystem's actors have been established, we offer a different perspective on the processes involved in quantum technological innovation in Spain.

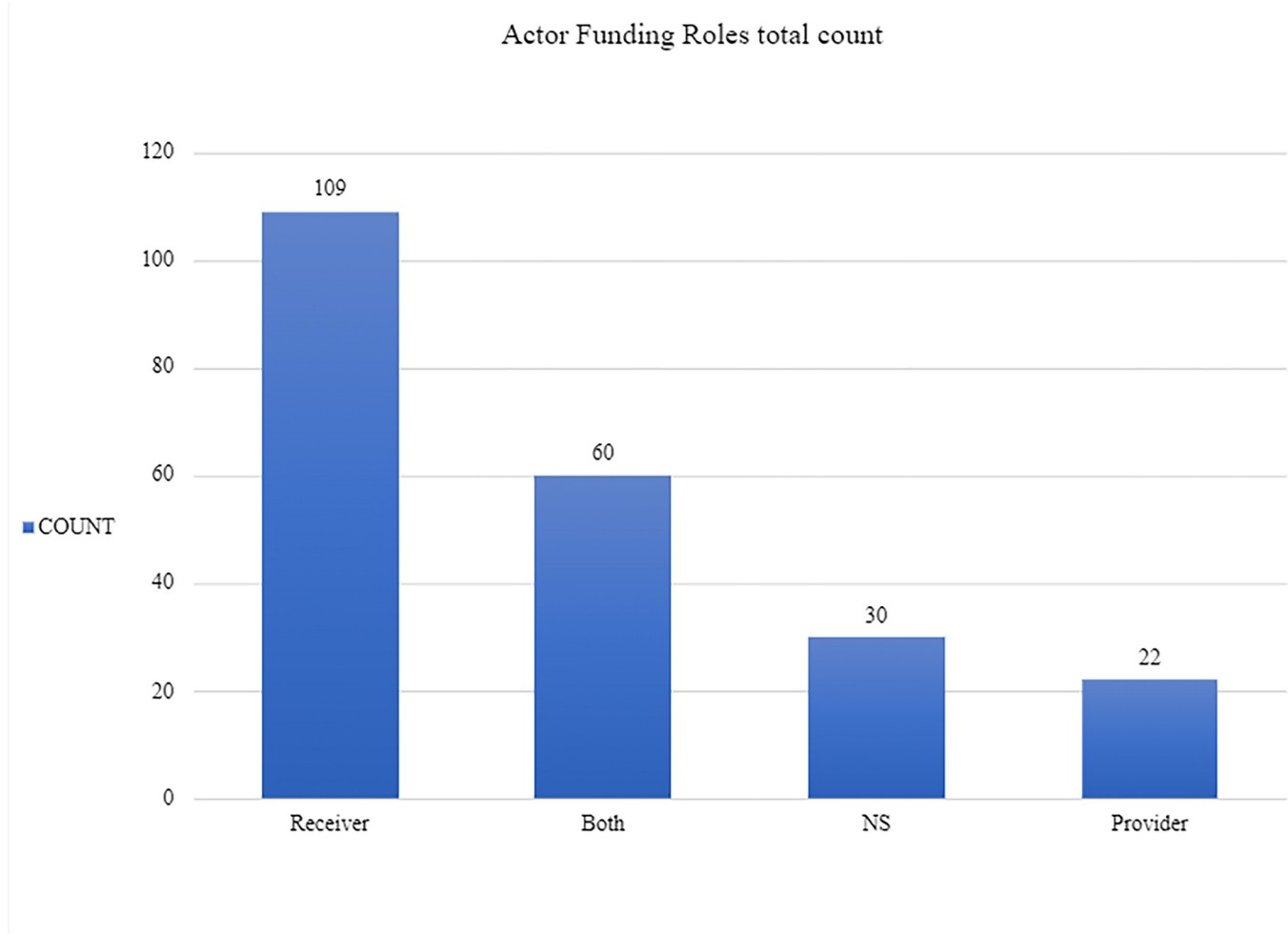

**Fig 6. Funding roles total count.**

The first notable finding is the determinant role that projects play in this ecosystem. This is signaled by the fact that they make up 1/4 of the total number of actors. This raises the question, therefore, of the contribution of projects to the development of quantum technologies and relatedly of the role played by projects in coordinating relationships between other parts of the ecosystem.

From our data, projects can be seen to have several common traits that indicate their role in the ecosystem. First, they do not have a physical location, unlike all the other actors. Second, they are always limited in time with a start and end date. Third, they tend to involve several other actors or at least two. Finally, they invariably fall under the category of both providing and receiving funding. All these indicators point out that projects function as intermediaries between funder and receiver institutions, thus acting as hubs to concentrate funding towards one specific avenue of innovation. In this sense, the projects may be understood at present as a key feature in defining the current ecosystem structure and its possible transformations over time. This is particularly significant because in current surveys and reports on quantum ecosystems, projects have been conflated with other types of actor thus obscuring the distinct roles they potentially play.

## Actor Funding Roles by percentage

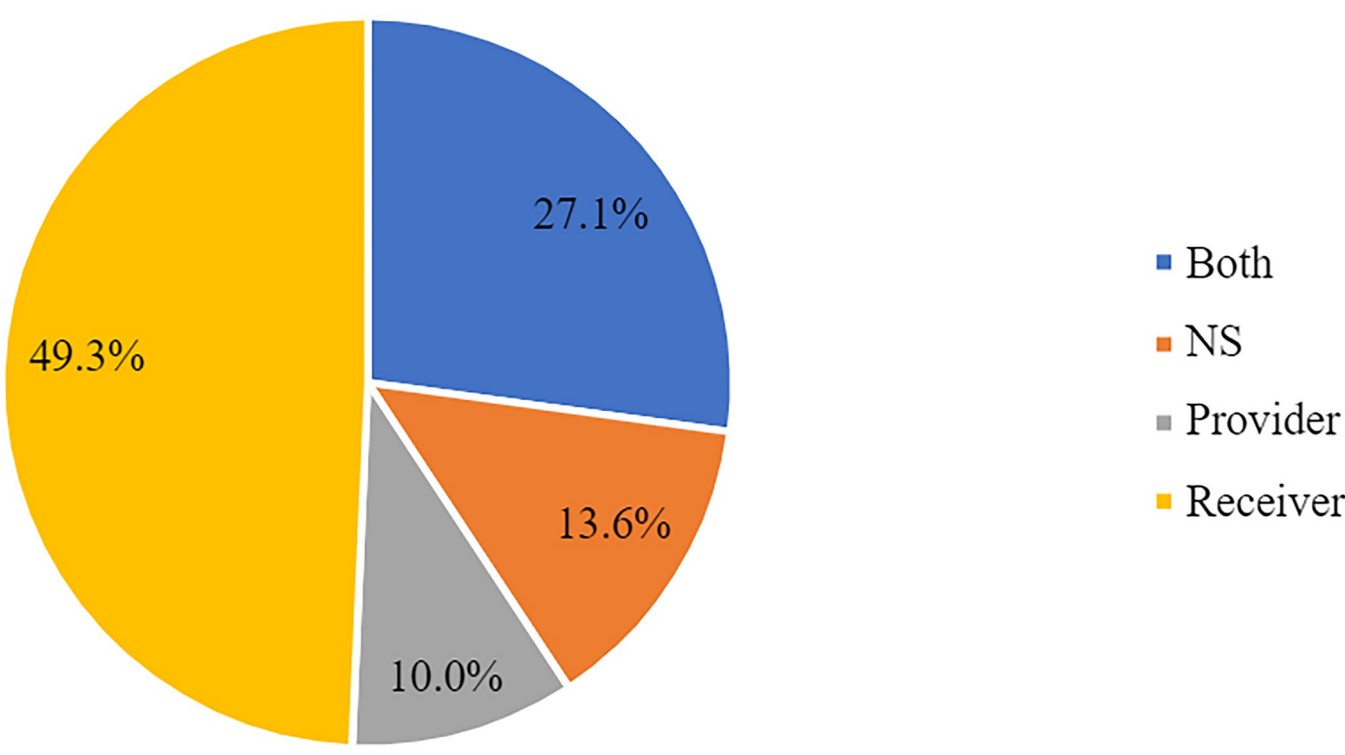

**Fig 7. Funding roles by percentage.**

The second notable finding is that most of the actors receive their funding from almost exclusively public institutions. The European Commission—as part of the European Union—clearly accounts for a large share of the invested capital, funding 43.9% of the total actors, followed by the Spanish and Catalan governments with 42% and 13.9% respectively. In contrast, private institutions such as the laCaixa Foundation or Kutxa Fundatioa only fund one single actor respectively. This suggests that quantum technological innovation in Spain is currently driven largely by the public sector since the capital for research is mainly provided by public institutions. Moreover, their funding tends to go through the aforementioned projects which is also relevant in analyzing the visibility of the public sector in innovation ecosystems and the wider sociopolitical reasons behind this 'screening' of public involvement.

The third notable finding is the prominence of research centres (both universities and independent institutes) in the category of Receivers which is already the most populated one. For instance, ICFO gathers funding from 25 different actors pointing out its enormous influence in the ecosystem. Consequently, further definition of its activities will be crucial in determining the state of the industry. Moreover, other research centres such as the BSC, UAB, DIPC, or UPV also participate in multiple projects. This suggests that innovation activity is still in—or dependent upon resources from—the research phase in most cases and the ecosystem is far from that of implementation or commercialization, a conclusion supported by the low number

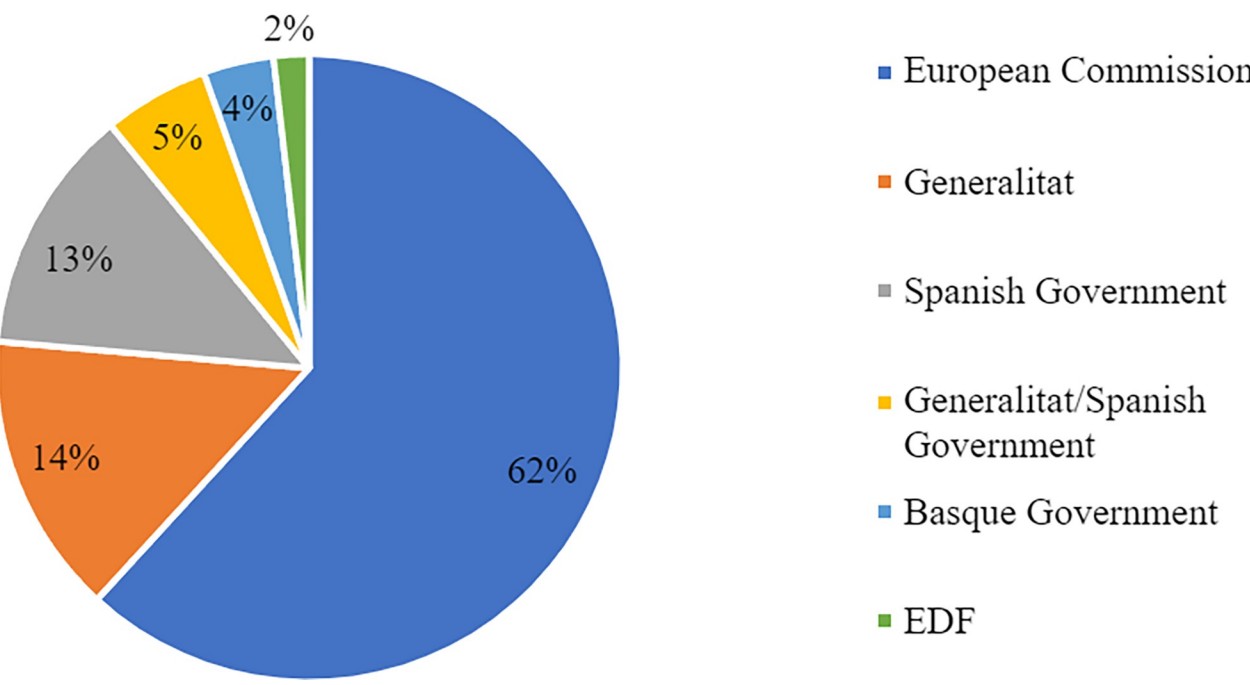

**Fig 8. Funding sources for projects by percentages.**

of Large Companies involved in it. This is significantly different impression compared to what has been called the 'hype' of quantum innovative activity.

There are other interesting trends shown by the findings such as the high concentration of actors in a few regional areas (Catalonia, Galicia, and the Basque country) which are also the headquarters of the national supercomputing initiatives, pointing out the potential symbiotic relationship between high-tech industries. In the same line, we observe that the technological subsector with more allocated funding is quantum computing which is consistent with the emphasis placed on the sector by the existing literature. Finally, there is a significant presence of publicly financed 'hubs' and 'ecosystems' which gives further credence to the innovation ecosystems theoretical approach.

## Section V: Conclusions

Current literature on the development of quantum technologies has done much to emphasize the number of active-interested parties related to the development and the implementation of quantum innovations. However, there has been an absence of attention to the particular connections between those listed parties and the character of those relations.

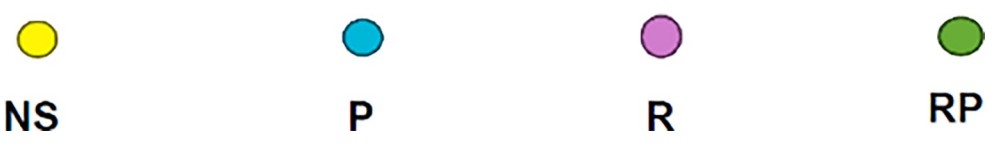

**Fig 9. Colour code for network analysis.**

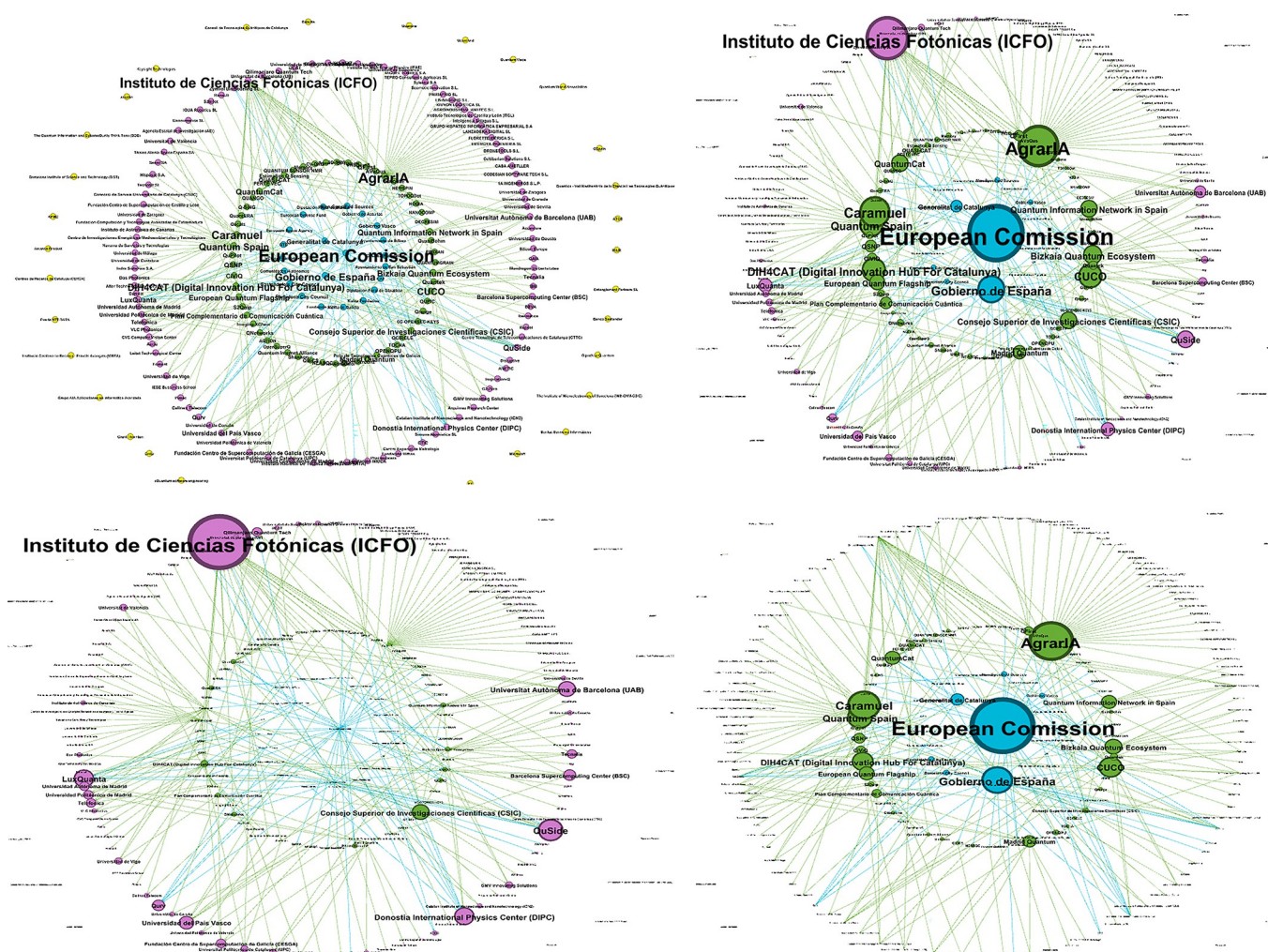

**Fig 10. a.** Overview of the Spanish Quantum Network. **b.** Network by Total degree. **c.** Network by In-degree. **d.** Network by Out-degree.

Drawing upon those institutions listed in current academic literature and private sector reports, this paper has taken one of the possible connections that typically exist between institutions in an innovation ecosystem—namely that of funding—to offer a new characterization of the quantum ecosystem in Spain.

This paper has found that with regards to this relationship, there are three key features of this ecosystem that are largely invisible in current literature. First, the significance and distinction of projects as actors in this ecosystem. Second, the strong degree to which funding both to and from projects comes from the public sector, particularly through institutions that make up the European Union. Third, the high degree of funding going from project to research centres, suggesting the relative importance of universities and research institutions relative to private sector companies in the ecosystem at this stage of its development.

Based on these findings, we suggest that these results may not be specific to the Spanish case but may well be a feature of quantum ecosystems in Europe more generally. Importantly, these results also suggest that to understand the characteristics and potential development of quantum technologies globally, further work on the specific relations between those actors in regions in the United States, China, and Israel is required.

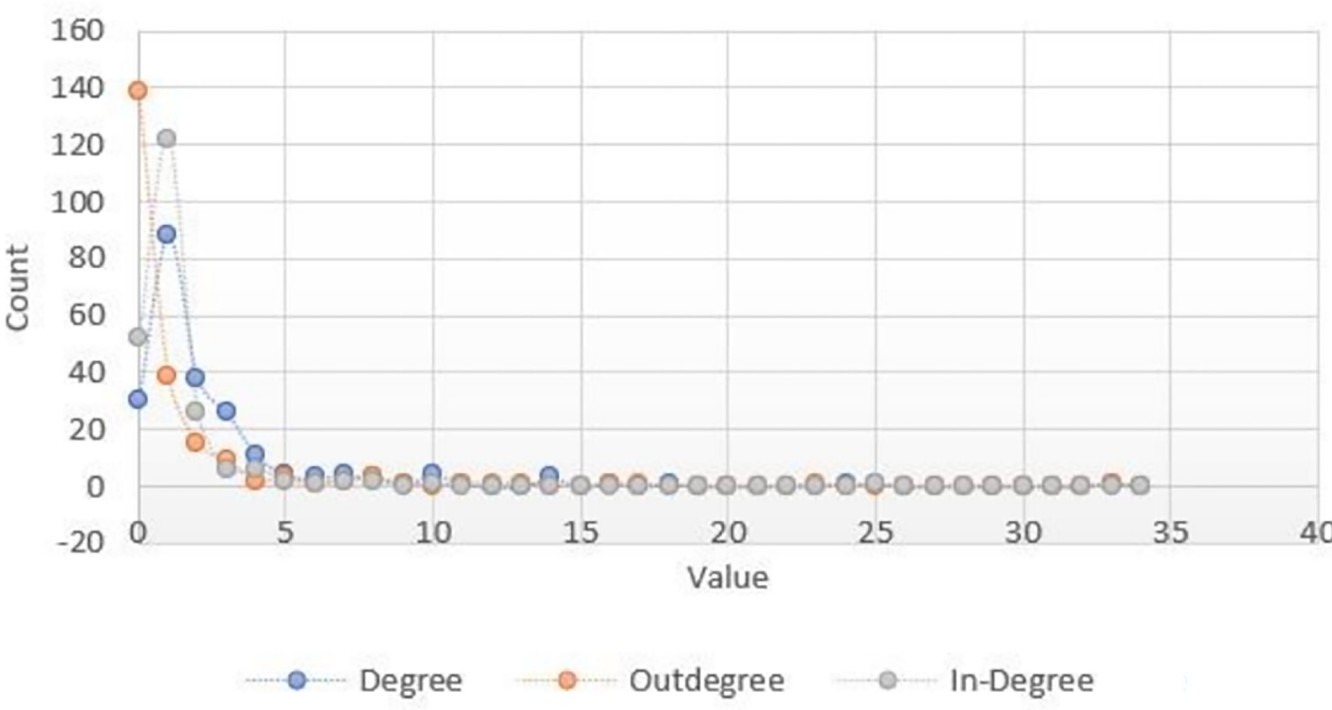

**Fig 11. Node breakdown by degree of centrality.**

## Supporting information

**S1 Table. Literature review breakdown.**
(DOCX)

**S2 Table. Actor breakdown.**
(DOCX)

## Author Contributions

**Conceptualization:** O. Jiménez Farías, Arnau Demergasso, Eoín Phillips.

**Data curation:** O. Jiménez Farías, Maryam Vaziri, Sergi Vives Rodón, Nelly Canessa, Eoín Phillips.

**Investigation:** O. Jiménez Farías, Arnau Demergasso, Eoín Phillips.

**Methodology:** Maryam Vaziri.

**Project administration:** O. Jiménez Farías, Eoín Phillips.

**Software:** Sergi Vives Rodón.

**Supervision:** Eoín Phillips.

**Validation:** Arnau Demergasso.

**Visualization:** Maryam Vaziri, Sergi Vives Rodón.

**Writing – original draft:** O. Jiménez Farías, Arnau Demergasso, Eoín Phillips.

**Writing – review & editing:** O. Jiménez Farías, Arnau Demergasso, Eoín Phillips.

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
