## [Decision Letter · Decision Letter 0]

20 Mar 2024

PONE-D-23-43053Visualising Quantum Innovation: A regional case studyPLOS ONE

Dear Dr. Jiménez Farías,

Thank you for submitting your manuscript to PLOS ONE. After careful consideration, we feel that it has merit but does not fully meet PLOS ONE’s publication criteria as it currently stands. Therefore, we invite you to submit a revised version of the manuscript that addresses the points raised during the review process.

**ACADEMIC EDITOR: **

Accepted after minor changes. 

The introduction provides a good, generalized background of the topic that quickly gives the reader an appreciation of the wide range of applications for this technology. This section helpfully explains the motivation for the research to current and potential funding agencies. However, to make the motivation clearer and to differentiate the paper some more from other applied papers, the author may wish to provide another sentence giving examples of some of the applications of this technology, along with appropriate references.

The objective is clearly defined in the last paragraph of introduction section and clearly mentions the flow of the paper. The experimental apparatus is quite standard, and is appropriate for the study, especially given that the focus of the paper is to develop a privacy preserving platform for industrial applications.

Author may wish to elaborate the problem formulation section and add some more details which will make the paper look more vital. I do not think any additional graphics are necessary. The author may also wish to give a more detailed discussion on blockchain and machine learning implementation.

The author may wish to mention why it is important to leverage blockchain to explain the motivation for his choice of specimens and accompany this with some references to other studies that demonstrate this importance. The authors are advised to add few latest citations, kindly read and cite: a) Smart Contract-Enabled Secure Sharing of Health Data for a Mobile Cloud-Based E-Health System. Appl. Sci. 2023, 13, 3970. https://doi.org/10.3390/app13063970, b) A deep learning approach for facial emotions recognition using principal component analysis and neural network techniques. The Photogrammetric Record, 37, 435–452.c) A secure framework for IoT-based smart climate agriculture system: Toward blockchain and edge computing" Journal of Intelligent Systems, vol. 31, no. 1, 2022, pp. 221-236. https://doi.org/10.1515/jisys-2022-0012

We look forward to receiving your revised manuscript.

Kind regards,

Mudassir Khan, Ph.D

Academic Editor

PLOS ONE

2. In your Methods section, please include additional information about your dataset and ensure that you have included a statement specifying whether the collection and analysis method complied with the terms and conditions for the source of the data.

4. Please amend the manuscript submission data (via Edit Submission) to include author Rodón, N. Canseca.

5. Please amend your authorship list in your manuscript file to include author Nelly Canessa Araujo.

Additional Editor Comments:

The introduction provides a good, generalized background of the topic that quickly gives the reader an appreciation of the wide range of applications for this technology. This section helpfully explains the motivation for the research to current and potential funding agencies. However, to make the motivation clearer and to differentiate the paper some more from other applied papers, the author may wish to provide another sentence giving examples of some of the applications of this technology, along with appropriate references.

The objective is clearly defined in the last paragraph of introduction section and clearly mentions the flow of the paper. The experimental apparatus is quite standard, and is appropriate for the study, especially given that the focus of the paper is to develop a privacy preserving platform for industrial applications.

Author may wish to elaborate the problem formulation section and add some more details which will make the paper look more vital. I do not think any additional graphics are necessary. The author may also wish to give a more detailed discussion on blockchain and machine learning implementation.

The author may wish to mention why it is important to leverage blockchain to explain the motivation for his choice of specimens and accompany this with some references to other studies that demonstrate this importance. The authors are advised to add few latest citations, kindly read and cite: a) Smart Contract-Enabled Secure Sharing of Health Data for a Mobile Cloud-Based E-Health System. Appl. Sci. 2023, 13, 3970. https://doi.org/10.3390/app13063970, b) A deep learning approach for facial emotions recognition using principal component analysis and neural network techniques. The Photogrammetric Record, 37, 435–452.c) A secure framework for IoT-based smart climate agriculture system: Toward blockchain and edge computing" Journal of Intelligent Systems, vol. 31, no. 1, 2022, pp. 221-236. https://doi.org/10.1515/jisys-2022-0012

Reviewers' comments:

Reviewer's Responses to Questions

**Comments to the Author**

1. Is the manuscript technically sound, and do the data support the conclusions?

Reviewer #1: Yes

Reviewer #2: Yes

2. Has the statistical analysis been performed appropriately and rigorously? 

Reviewer #1: Yes

Reviewer #2: Yes

3. Have the authors made all data underlying the findings in their manuscript fully available?

Reviewer #1: Yes

Reviewer #2: Yes

4. Is the manuscript presented in an intelligible fashion and written in standard English?

Reviewer #1: Yes

Reviewer #2: Yes

5. Review Comments to the Author

Reviewer #1: The article is based on country specific. The author present a literature review that considers a set of reports by several organizations.

Author should mention the name of organization too.

All the tables and figures should be described within the article.

Reviewer #2: this research has collected comprehensive data and conducted extensive analysis regarding the quantum technological innovation Spain. the results of this research may be expanded to other regions in Europe, as well as help in the potential development of quantum technologies globally.

6. PLOS authors have the option to publish the peer review history of their article (what does this mean?). If published, this will include your full peer review and any attached files.

Reviewer #1: **Yes: **Dr Mahtab Alam

Reviewer #2: No

---

## [Author Response · Author response to Decision Letter 0]

8 May 2024

Dear Editor

We are very happy to know that our paper “Visualizing Quantum Innovation: A regional Case Study” meets the merits for publication in PLOS One while requiring minor revisions. 

 In what follows we, present a point-by-point response to the editor and reviewer’s comments and suggestions as well an important modification to the authors list that that has been decided internally.

Academic editor: “The introduction provides a good, generalized background of the topic that quickly gives the reader an appreciation of the wide range of applications for this technology.”

Authors: We would like to thank the editor for appreciating this value of our article

Academic editor: “However, to make the motivation clearer and to differentiate the paper some more from other applied papers, the author may wish to provide another sentence giving examples of some of the applications of this technology, along with appropriate references.” 

Authors: We have added 8 new citations including those that the editor kindly recommend in the introductory sentence: “Potential application of quantum technology range drug discovery (Santagati et al, 2024) , financial modelling (Herman et al, 2023) , weather forecasting (Tennie et al, 2023), cybersecurity (Neil, 2023) , machine learning algorithms (Biamonte et al, 2017, Khan et al, 2022) and Bloch chain (Wang et al, 2022, Chinnasamy et al, 2023)”

Academic editor: The objective is clearly defined in the last paragraph of introduction section and clearly mentions the flow of the paper. The experimental apparatus is quite standard, and is appropriate for the study, especially given that the focus of the paper is to develop a privacy preserving platform for industrial applications. 

Authors: We thank the editor 

Reviewer #1: The article is based on country specific. The authors present a literature review that considers a set of reports by several organizations.

Author should mention the name of organization too.

All the tables and figures should be described within the article.

Authors: We have included the name of the organizations in the section “Structure of the Study” in the sentence: “we present a literature review that considers a set of reports by several organizations, like the World Economic Forum, Ametic, McKinsey, Quantum Flagship, IBM, IQM, among others,”

Reviewer #2: this research has collected comprehensive data and conducted extensive analysis regarding the quantum technological innovation Spain. the results of this research may be expanded to other regions in Europe, as well as help in the potential development of quantum technologies globally.

Authors: We thank reviewer #2 for her/his encouraging comment.

Yours sincerely,

Osvaldo Jiménez Farías and Eóin Phillips

On behalf of the authors.

---

## [Editor Report · Decision Letter 1]

27 May 2024

Visualising Quantum Innovation: A regional case study

PONE-D-23-43053R1

Dear Dr.Osvaldo Jiménez Farías

We’re pleased to inform you that your manuscript has been judged scientifically suitable for publication and will be formally accepted for publication once it meets all outstanding technical requirements.

Kind regards,

Mudassir Khan, Ph.D

Academic Editor

PLOS ONE

Additional Editor Comments (optional):

Thanks to the authors for the detailed response and additions. I read through the comments and skimmed the revised PDF, and the updates significantly improved the paper. I would be happy to recommend this paper for publication.
---

## [Editor Report · Acceptance letter]

30 May 2024

PONE-D-23-43053R1 

PLOS ONE

Dear Dr. Jiménez Farías, 

I'm pleased to inform you that your manuscript has been deemed suitable for publication in PLOS ONE. Congratulations! Your manuscript is now being handed over to our production team.

Kind regards, 

on behalf of

Dr. Mudassir Khan 

Academic Editor

PLOS ONE